# Molecular and Genomic Analysis of the Virulence Factors and Potential Transmission of Hybrid Enteropathogenic and Enterotoxigenic *Escherichia coli* (EPEC/ETEC) Strains Isolated in South Korea

**DOI:** 10.3390/ijms241612729

**Published:** 2023-08-12

**Authors:** Woojung Lee, Soohyun Sung, Jina Ha, Eiseul Kim, Eun Sook An, Seung Hwan Kim, Soon Han Kim, Hae-Yeong Kim

**Affiliations:** 1Division of Food Microbiology, National Institute of Food and Drug Safety Evaluation, Ministry of Food and Drug Safety, Cheongju 28159, Republic of Korea; woojungluv79@korea.kr (W.L.); sungsoohyun@korea.kr (S.S.); jinaha423@korea.kr (J.H.); jupiter1255@korea.kr (E.S.A.); betterkim@korea.kr (S.H.K.); 2Institute of Life Sciences & Resources, Department of Food Science and Biotechnology, Kyung Hee University, Yongin 17104, Republic of Korea; eskim89@khu.ac.kr

**Keywords:** enteropathogenic *E. coli*, enterotoxigenic *E. coli*, whole-genome sequencing, virulence factor, plasmid, bacteriophages

## Abstract

Hybrid strains *Escherichia coli* acquires genetic characteristics from multiple pathotypes and is speculated to be more virulent; however, understanding their pathogenicity is elusive. Here, we performed genome-based characterization of the hybrid of enteropathogenic (EPEC) and enterotoxigenic *E. coli* (ETEC), the strains that cause diarrhea and mortality in children. The virulence genes in the strains isolated from different sources in the South Korea were identified, and their phylogenetic positions were analyzed. The EPEC/ETEC hybrid strains harbored *eae* and *est* encoding *E. coli* attaching and effacing lesions and heat-stable enterotoxins of EPEC and ETEC, respectively. Genome-wide phylogeny revealed that all hybrids (*n* = 6) were closely related to EPEC strains, implying the potential acquisition of ETEC virulence genes during ETEC/EPEC hybrid emergence. The hybrids represented diverse serotypes (O153:H19 (*n* = 3), O49:H10 (*n* = 2), and O71:H19 (*n* = 1)) and sequence types (ST546, *n* = 4; ST785, *n* = 2). Furthermore, heat-stable toxin-encoding plasmids possessing *estA* and various other virulence genes and transporters, including *nleH2, hlyA*, *hlyB*, *hlyC*, *hlyD*, *espC*, *espP*, phage endopeptidase Rz, and phage holin, were identified. These findings provide insights into understanding the pathogenicity of EPEC/ETEC hybrid strains and may aid in comparative studies, virulence characterization, and understanding evolutionary biology.

## 1. Introduction

*Escherichia coli* is a diverse species of enteric bacteria implicated in several severe clinical outcomes, including diarrhea, acute inflammation, hemorrhagic colitis, urinary tract infections, septicemia, and neonatal meningitis. *E. coli*, mostly the benign members of the commensal gut microbiota, and some strains that have acquired specific virulence cause diarrheagenic and extraintestinal illnesses in humans and other animals [1,2]. Diarrheagenic *E. coli* (DEC) causes 30–40% of acute diarrhea episodes in children aged < 5 years in developing countries [3]. According to the 2010 World Health Organization (WHO) Global Burden of Foodborne Diseases report, over 111 million illnesses and nearly 63,000 deaths annually are caused by DEC [4].

DEC isolates are subdivided into a number of pathotypes based on the presence of specific virulence characteristics that are directly associated with disease development [5,6,7,8]. The Global Enteric Multicenter Study (GEMS), which was a case-control epidemiological investigation of the causes of diarrhea among children younger than 5 years of age in 7 different countries in South Asia and sub-Saharan Africa over the course of three years, found that multiple diarrheagenic pathotypes of *E. coli* were the leading cause of severe childhood diarrhea [9,10]. Heat-stable enterotoxic (ST-ETEC) and enteropathogenic *E. coli* (EPEC) isolates were associated with an increased risk of mortality among the *E. coli* strains examined in the GEMS [10]. Furthermore, the study also highlighted the key role of the host environment in the development of the disease—even though the diarrheagenic pathotypes were isolated from children with and without diarrhea in the GEMS [10,11,12], the presence of known pathotype-associated diarrheagenic virulence factors caused diarrhea [13].

EPEC strains are categorized as either typical or atypical according to whether or not they have the *E. coli* adherence factor plasmid (EAF) expressing the *bfpA* gene. This gene is responsible for the production of bundle-forming pili. Typical EPEC (tEPEC) have both the locus of enterocyte effacement (LEE) region for attaching and effacing lesions (*eaeA*) and *bfpA*, whereas atypical EPEC (aEPEC) do not contain *bfpA* [5,14]. aEPEC isolates show similar prevalence among healthy adults and children with diarrhea and are also identified in healthy adults [11,15,16]; however, some aEPEC isolates have been associated with diarrheal outbreaks [17,18,19,20,21]. ETEC colonizes the small intestine and causes watery diarrhea, known as traveler’s diarrhea, in humans by producing plasmid-encoded heat-stable (ST) or heat-labile enterotoxins (LT) [5]. In the isolates of these pathovars, these canonical virulence factors are frequently encoded by plasmids or other mobile elements [5,6].

Mobile genetic elements (MGEs) such as insertion sequences, phages, and plasmids usually carry several virulence markers that allow for the horizontal transfer of virulence genes and subsequent formation of hybrid pathotypes [5,22,23,24,25]. With technological advances, genetic markers of different pathotypes in hybrid *E. coli* strains have been identified, providing a better understanding of the genomic and virulence mechanisms of DEC [26].

The *E. coli* O104:H4 strain is a widely known case of this occurrence; in 2011, it was responsible for a significant outbreak of acute gastroenteritis and HUS in Germany [27]. In addition, this strain carried a plasmid that included genes for aggregative adherence fimbriae (AAF), which are the proteins that are responsible for aggregative adherence in EAEC [28,29,30]. The production of Stx2 is a characteristic that is unique to the STEC pathotype. Furthermore, hybrids of the EPEC and ETEC strains (EPEC/ETEC) have recently been reported in Africa, India, Iran, and Mexico, some of which are associated with watery and bloody diarrheal diseases in humans [9,10,31,32,33]. Additionally, EPEC and some ETEC strains have been linked to increased death rates, further emphasizing the importance of DEC in the global burden of diarrheal diseases [34].

Few studies have reported genome-based characteristic analyses of EPEC/ETEC hybrid strains isolated in the South Korea. In this study, we aimed to investigate the genomes of EPEC/ETEC hybrid strains to identify virulence factors from both EPEC and ETEC pathovars. We also figured out where the discovered strains fit in a large group of pathogenic *E. coli* strains that included all of the major pathotypes. Real-time polymerase chain reaction (PCR) and whole-genome sequencing (WGS) were used for investigation into the molecular features of these strains. To determine the evolutionary locations of these hybrids, a phylogenetic analysis was carried out. Genome-wide phylogenetic analysis revealed that these hybrids were closely related to certain EPEC strains. Through genome-based characterization, we confirmed that the EPEC/ETEC hybrid strains are likely EPEC strains that have acquired ETEC virulence genes via plasmids. On the basis of our research, we explored the possible effects that the hybrid *E. coli* strains detected may have on the health of the general public.

## 2. Results

### 2.1. Genome Assemblies of EPEC/ETEC Hybrids

Six EPEC/ETEC hybrid *E. coli* isolates from the South Korea were sequenced—four isolates (MFDS1009724, MFDS1013798, MFDS1015962, and MFDS1016213) had one chromosome and one plasmid, and two (MFDS101796 and MFDS1017708) had one chromosome and two plasmids. The genomic characteristics of the hybrid EPEC/ETEC isolates are summarized in Table 1. The length of the genomes of these different isolated strains varied from 5,132,531 bp to 5,434,752 bp, and the coverage varied from 251 to 643×. Additionally, these strains had 50.7–50.9% G + C contents, 5120–5756 coding DNA sequences (CDSs), 93–99 tRNA genes, and 22 rRNA genes.

### 2.2. In Silico Identification of Virulence Genes

Using real-time PCR, pathogenic *E. coli* strains were initially screened. The hybrid EPEC/ETEC isolates contained the genes *eaeA* and *est*, which code for intimin and heat-stable enterotoxin, respectively. Following that, the mapping of virulence genes was carried out to discover a variety of different virulence factors that have been discovered in the hybrid genomes of EPEC and ETEC (Figure 1). Multiple virulence factors have been linked to the pathogenesis of *E. coli*, including LEE-related genes and enterotoxins, as well as the LEE-locus-encoded type III secretion system (TTSS), LEE-encoded TTSS effectors, and heat-stable enterotoxins. Appendix A details the outcomes of the virulence mapping of genes in these hybrid genomes. We identified LEE-encoding genes that are associated with the pathogenicity of EPEC. A comparison of the six hybrid EPEC/ETEC isolates with EPEC strains (E2348_69 [35] and E110019 [36]) and ETEC strains (214-4 [37] and H10407 [38]) isolated from adult feces is shown in Figure 2. Considering E2348_69 and E110019 as positive controls, we demonstrated that all EPEC/ETEC strains contained the LEE region.

### 2.3. Serotyping and Sequence Types of the EPEC/ETEC Hybrids

*E. coli* O and H antigens were serotyped using specific somatic (O1–O181) antisera and the WGS-based SerotypeFinder v2.0. The sequence type (ST) was identified using the *E. coli* and *Shigella* species whole genome multi-locus sequence typing (wgMLST) database. Serotyping and sequencing results for the six hybrid EPEC/ETEC isolates are summarized in Table 2. Based on in vitro and in silico serotyping, the six hybrid *E. coli* strains belonged to three distinct O:H serogroups [O153:H19 (*n* = 3), O49:H10 (*n *= 2), and O71:H19 (*n* = 1)]. The hybrid strains comprised two sequence types: [ST546 (*n* = 4) and ST785 (*n* = 2)].

### 2.4. Comparative Analysis of Heat-Stable Toxin-Encoding Plasmids

To investigate the plasmid-mediated heat-stable enterotoxin (*est*) horizontal gene transfer of hybrid EPEC/ETEC isolates, an analysis of plasmid-associated sequences was conducted using PlasmidFinder 2.1. The analysis revealed the presence of two plasmid replicon sequences (IncFIl and IncFIA) belonging to known incompatibility (Inc) groups in the EPEC/ETEC genomes (Table 3 and Figure 3). As shown in Appendix A, *estA* was located in the same contig as IncFIl. The mobile genetic elements surrounding *estA* are shown in Appendix A. Furthermore, heat-stable toxin-encoding plasmids carried not only *estA* but also various virulence genes and transporters, such as *nleH2*, *hlyA*, *hlyB*, *hlyC*, *hlyD*, *espC*, *espP*, phage endopeptidase Rz, and phage holin.

In addition, based on the analysis of phages in heat-stable toxin-encoding plasmid genomes, the 62.1-kb phage Stx2_c_1717 (accession number NC_0011357) or the 57.9-kb *Enterobacteria* phage Entero_BP_4795 (NC_004813) were found on the plasmids. However, the 39-kb *Enterobacteria* phage Shigel_Sf6 (accession number NC_005344), 103.4-kb *Salmonella* phage Salmon_SJ46 (accession number NC_031129), and 115.2-kb *Enterobacteria* phage Escher_RCS47 (NC_042128) were found only in the plasmid genomes of MFDS1013798 or MFDS1016213. Virulence genes and transporter sequences were confirmed in phage sequence regions corresponding to “intact”.

### 2.5. Analysis of Phylogenetic and Population Structure

A phylogenetic analysis was performed on a total of 187 strains to identify the genomic links between EPEC/ETEC hybrids and other pathogenic *E. coli* isolates. This study employed the genomes of 14 STEC isolates, 46 ETEC isolates, 72 EPEC isolates, 18 EAEC isolates, and 10 EIEC isolates. An analysis of phylogenetic trees showed that the six hybrids were closely linked to specific EPEC strains, which suggests that they may have acquired ETEC virulence genes during the process of their emergence (Figure 4A). Furthermore, RhierBAPSs were used to characterize the population structures of the 187 genome datasets, which separated the genome datasets into five basic sequence clusters (hierarchical level 1 of the Bayesian analysis of population structure, called BAPS). These were further subdivided into a total of 27 lineages (BAPS level 2). The results revealed that the six EPEC-related hybrid strains could be classified into two categories (level 1 and level 2).

## 3. Discussion

EPEC/ETEC hybrids are found in various sources, including humans, animals, food, and water, some of which are associated with watery and bloody diarrheal diseases. The EPEC/ETEC hybrid strain was first isolated in India from a 6-month-old child presenting with acute watery and bloody diarrhea, a more severe symptom in India [31]. Here, we report six EPEC/ETEC hybrid strains identified in the South Korea between 2016 and 2020. Genomic and phylogenetic analyses demonstrated that the EPEC/ETEC hybrid isolates were genomically related to EPEC and appeared to have acquired ETEC virulence genes via plasmids. Furthermore, comparative plasmid analysis of EPEC/ETEC hybrid strains revealed several genes responsible for specific genomic features. The molecular characterization of all the pathogenic *E. coli* strains using real-time PCR and serotyping was consistent with that of WGS analysis. Based on virulence markers, EPEC strains are classified as typical or atypical according to the presence or absence of the EAF plasmid that carries *bfpA*, which encodes bundle-forming pili (BFP). Both *eaeA* and *bfpA* were identified in tEPEC, while aEPEC strains lacked *bfpA*. It has been reported that aEPEC causes endemic, epidemic, and persistent diarrhea in humans and farm animals [14], some of which have been associated with acute disease in humans [39]. Taken together, we confirmed that the EPEC/ETEC hybrid strains identified in the South Korea are aEPEC strains that lack the gene encoding BFP.

All six EPEC/ETEC hybrid strains contained chromosomally encoded virulence genes of EPEC, such as intimin and TTSS of LEE, and other non-LEE-encoded effectors. The EPEC/ETEC hybrid strains also harbored a plasmid-encoded ETEC virulence gene (*est*), which encodes a heat-stable enterotoxin. Furthermore, the phylogenetic tree analysis revealed that the six hybrids were closely related to EPEC. These results demonstrate that the EPEC/ETEC hybrid strains isolated in the South Korea harbor pathotype-specific virulence factors for EPEC and ETEC and are likely EPEC strains that have acquired ETEC virulence genes via plasmids.

In addition, heat-stable toxin-encoding plasmids were found to carry *estA* and various virulence genes and transporters, such as *nleH2*, *hlyA*, *hlyB*, *hlyC*, *hlyD*, *espC*, *espP*, Hok/Gef-like protein, phage holin, and phage endopeptidase Rz. The EPEC/ETEC hybrid strains all contained *estA*, *nleH2*, Hok/Gef-like protein, and phage endopeptidase Rz, whereas *hlyA*, *hlyB*, *hlyC*, *hlyD*, and *espP* were encoded in all the EPEC/ETEC hybrid strains except MFDS1016213. In particular, phage holin was present only in MFDS1013798.

The *nleH2* gene has been linked to the inhibition of both innate inflammatory and cell death signaling pathways, as well as the alteration of the cytoskeletal composition of attaching and effacing lesions during EPEC infection [40,41]. In addition, *espC* and *espP*, the serine protease autotransporters of Enterobacteriaceae, have been reported to play a role in regulating pore formation and cytotoxicity mediated by the TTSS during EPEC or EHEC infection [42,43]. In addition to this, α-hemolysin is a component of an operon that contains four genes: *hlyA* (which encodes the functional toxin HlyA), *hlyC* (which is responsible for the post-translational modification of HlyA), *hlyB*, and *hlyD* (which are responsible for the transport of HlyA across the inner membrane) [44,45]. Previous research [46] has demonstrated that a pore-forming cytolysin functions as a virulence factor in intestinal and extraintestinal pathogenic *E*. *coli* strains.

In particular, phages are responsible for encoding some virulence factors. Numerous bacteriophages bear virulence genes encoding proteins that play crucial roles in the pathogenesis of bacteria [47]. Through a process referred to as phage lysogenic conversion, bacteriophages that encode virulence factors have the ability to convert the non-pathogenic strain of bacteria that they infect into either a virulent strain or a strain that shows increased levels of virulence. Specifically, phage holin and endolysin are necessary components for the lysis of the host cell [48]. Holins are the proteins that are responsible for creating holes in the cytoplasmic membrane. These holes act as transport channels for endolysins, which are the enzymes responsible for digesting the peptidoglycan layer. In addition to this, Rz/Rz1-like proteins frequently operate as accessory proteins that boost the activity of endolysin [49].

This study shows the transfer of virulence genes in South Korean-isolated EPEC/ETEC hybrid strains. The results of this study highlight that WGS is a powerful tool for analyzing bacterial genomes in the presence of regions of MGEs, such as phages and plasmids. In addition, the genomic information that was gathered throughout the course of this research has the potential to make a substantial contribution to an improved comprehension of the genetic properties of hybrid *E. coli* strains. Additional research is required to study the genomic and transcriptome properties of EPEC/ETEC hybrid strains that were isolated in the Korea from a variety of ecological and geographical sources.

## 4. Materials and Methods

### 4.1. Bacterial Strains and Serotyping

Pathogenic *E. coli* isolates (*n* = 1025) were isolated from animal food (beef, pork, chicken, and duck) (*n* = 649), animal feces (cattle, pigs, and poultry) (*n* = 290), vegetables such as salad (*n* = 70), and other (*n* = 16) during 2016 to 2020 in the Korea and distributed by the Korean Culture Collection for Foodborne Pathogens (Ministry of Food and Drug Safety, Cheongju, Republic of Korea). The geographic distributions of the strains were as follows: Gyeonggi-do (*n* = 281), Gangwon-do (*n* = 47), Chungcheong-do (*n* = 142), Jeolla-do (*n* = 491), and Gyeongsang-do (*n* = 64). Subcultures of blue-green *E. coli* colonies grown on BCIG (5-bromo-4-chloro-3-indolyl-D-glucuronide) agar (Oxoid, Hampshire, UK) were incubated at 37 °C for 18–24 h before being transferred to a new plate of Tryptic Soy Agar (Oxoid, UK). VITEK MS (BioMerieux Inc., Marcy-l’Etoile, France) was used to identify the isolates. Agglutination of the bacteria with particular somatic antisera (O1 to O181) was performed at the Laboratorio de Referencia de *E. coli* (LREC), located in Lugo, Spain, in order to discover variants of somatic (O) antigens [50,51,52]. This allowed for the determination of the serotype.

### 4.2. PCR-Based Identification of Hybrid Strains

DNA extraction from the bacterial cultures using automated equipment (EZ1 Advanced XL, Qiagen, Germantown, MD, USA) was performed according to the manufacturer’s recommendations and used as the DNA template. Real-time PCR was performed using a PowerCheck^TM^ 20/15 Pathogen Multiplex Real-time PCR kit (Kogen Biotech Co., Ltd., Seoul, Republic of Korea) to detect virulence genes in different DEC pathogens, including *VT1* and *VT2* (STEC); *bfpA* and *eaeA* (EPEC); *LT*, *STh* and *STp* (ETEC); *aggR* (EAEC); *ipaH* (EIEC). Amplification was performed using an ABI 7500 Fast Real-time PCR system (Applied Biosystems, Waltham, MA, USA) with the following thermal conditions: 1 cycle at 50 °C for 2 min, 1 cycle at 95 °C for 10 min, and 40 cycles at 95 °C for 15 s and 60 °C for 1 min.

### 4.3. Genome Sequencing, Assembly, and Annotation

Genomic DNA was extracted using the MagListo^TM^ 5M Genomic DNA Extraction Kit for cells and tissues (Bioneer, Daejeon, Republic of Korea) according to the manufacturer’s protocol. The gDNA concentration was determined with a Qubit^TM^ 3.0 Fluorometer and a Qubit double-stranded DNA (dsDNA) high-sensitivity (HS) assay kit (Thermo Fisher Scientific, Inc., Waltham, MA, USA). A sequencing library was prepared using a Nextera DNA Flex Library Prep Kit (Illumina, San Diego, CA, USA). The amplified library was assessed for quantity and quality using a Bioanalyzer 2100 instrument (Agilent Technologies, Waldbronn, Germany) with a HS DNA chip. Sequencing was performed using a MiSeq sequencing system (Illumina) and MiSeq Reagent Kit v3 (600 cycles). Hybrid genome assembly was performed using additional long-read sequence data obtained using PacBio Sequel (Pacific Bioscience, Menlo Park, CA, USA) to obtain high-quality data to determine the complete genome sequence. Hybrid assembly of raw FASTQ PacBio long-read sequence data and Illumina MiSeq short-read FASTQ sequence data was performed using Unicycler (v0.4.9, https://github.com/rrwick/Unicycler (accessed on 1 January 2023); default options). The assembled genome was annotated using the Rapid Annotation with Subsystem Technology toolkit as implemented in the PATRIC annotation web service (v3.6.12).

### 4.4. DNA Sequence and Bioinformatics Analysis

The virulence factors were identified using a virulence factor database [53]. Toxigenic and pathogenic factors in the human intestine required for EPEC pathogenesis, such as LEE-encoded virulence factors, were identified using a Basic Local Alignment Search Tool (BLAST v2.14.0). Mobile genetic elements were identified using MobileElementFinder (https://cge.cbs.dtu.dk/services/MobileElementFinder/) (accessed on 22 February 2023) [54]. wgMLST was carried out using BioNumerics (version 8.0; Applied Maths, Sint-Martens-Latem, Belgium), and the software incorporated 17,380 loci from *E. coli* and Shigella into its analysis. In addition, the web-based serotyping application known as SerotypeFinder 2.0 [55] was utilized in order to make predictions regarding the antigen profiles of different *E. coli* strains.

### 4.5. Identification of Plasmid-Associated Sequences

PlasmidFinder 2.1 was utilized in order to investigate the plasmid characteristics of entire genomes [56]. The threshold for identification was set to 95%, and the minimum coverage for identification was set to 60%. The finding of replicon sequences from multiple different known plasmid incompatibility (Inc) groups served as the foundation for the identification. Using the Rapid Annotation with Subsystem Technology tools integrated with the PATRIC annotation web service (v3.6.12), the retrieved plasmid sequences were annotated to detect virulent genes.

### 4.6. Prophage Prediction and Analysis

The PHAge Search Tool Enhanced Release (PHASTER) [57] was used to identify bacteriophage sequences in the unique genome sequence. PHASTER was used to predict putative prophage regions as “intact (score > 90),” “questionable (score 70–90),” and “incomplete (score < 70)” based on their sizes, similarity to known phages, and the presence of phage-like and phage cornerstone genes (for example, “capsid”, “head”, “plate”, “tail”, “coat”, “portal”, and “holin”) in the identified phage region of the heat-stable toxins-encoding plasmid genome. Using the RAST toolbox from the PATRIC genome annotation web service (v3.6.12), virulence genes in the recovered prophage sequences were annotated.

### 4.7. Phylogenetic and Population Structure Analysis

Comparative genomic analysis was performed on six EPEC/ETEC hybrid strains isolated from the South Korea, and 187 pathogenic *E. coli* that included 160 strains isolated from various types of food and different environments in the South Korea and 27 other pathogenic *E. coli* strains are available from the National Center for Biotechnology Information (NCBI). The bacterial pan-genome analysis (BPGA) tool (v1.3; default settings) was used to look at the pan-genome. For clustering, the USEARCH tool was used, and the cut-off number was set at 95% sequence identity. The neighbor-joining method was used to put the phylogenetic tree into groups, and Interactive Tree of Life (iTOL) v6 was used to show the groups. RhierBAPs were used to investigate the organization of the population [58].

## 5. Conclusions

The present study reports the virulence profiles of the EPEC/ETEC hybrid strains isolated in the South Korea. The virulence markers that are identified in EPEC and ETEC pathotypes have been demonstrated to be carried by these plasmids using a genome-based characterization. Additionally, we identified the adherence factor intimin, a type III secretion system, and several secreted proteins encoded by the LEE pathogenicity island. Moreover, heat-stable toxin-encoding plasmids carry *estA* and various virulence genes and transporters. Our results emphasize that the EPEC/ETEC hybrid strains isolated in the South Korea, which contained *est* and various virulence genes, may be more dangerous to humans than EPEC or ETEC alone. In other words, it is suggestive that the establishment of hybrid DEC strains has serious implications for public health and should be taken into account in patient management and epidemiological surveillance.

## Figures and Tables

**Figure 1 ijms-24-12729-f001:**
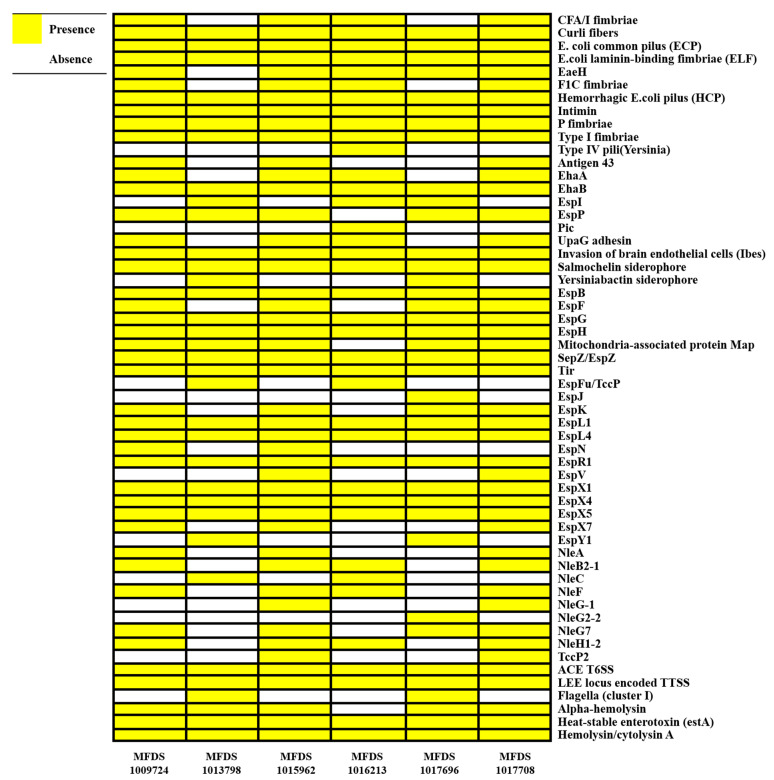
The heatmap of virulence factors across the hybrid EPEC/ETEC strains constructed using the gplot (v3.1.3) package in R (v4.1.3). Heatmap showing the presence/absence of virulence factors (*y*-axis) within the hybrid EPEC/ETEC isolates identified in this study (*x*-axis). Yellow indicates the presence of virulence genes; white indicates the absence of virulence genes.

**Figure 2 ijms-24-12729-f002:**
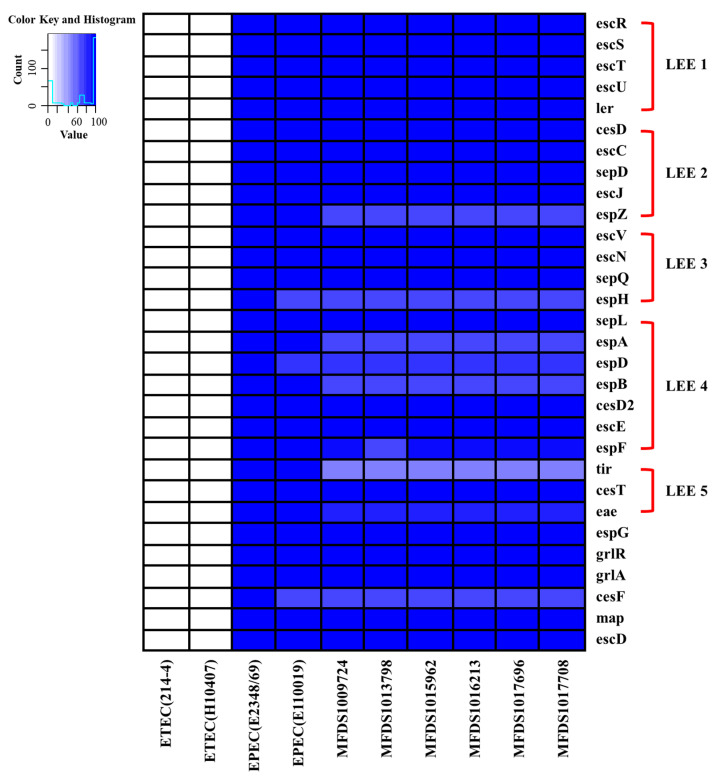
The heatmap of the protein-encoding genes within the LEE region of EPEC isolates E2348/69 and E110019, ETEC isolates H10407 and 214-4, and EPEC/ETEC genomes analyzed in this study. The Heatmap shows the presence/absence of the protein-encoding genes (*y*-axis) within the hybrid EPEC/ETEC isolates identified in this study (*x*-axis). The presence of virulence genes is shown in blue, and their absence is shown in white. A heatmap was produced using the gplot (v3.1.3) package in R (v4.1.3).

**Figure 3 ijms-24-12729-f003:**
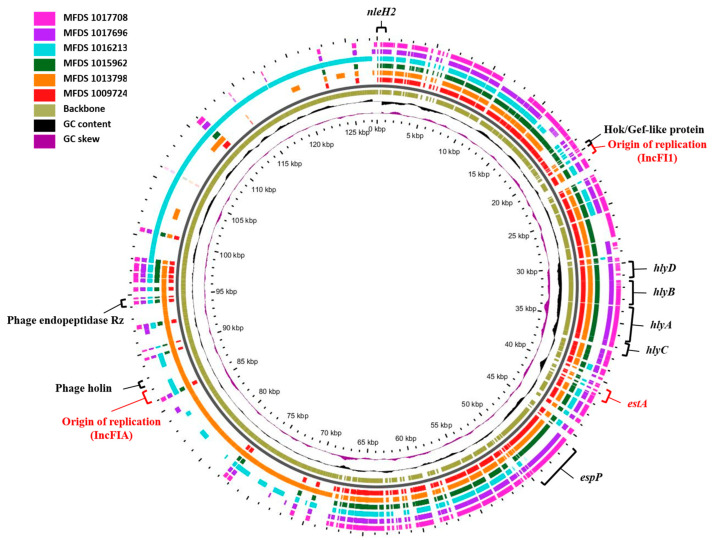
Comparative analysis of heat-stable toxin-encoding plasmids. The pangenome shows the heat-stable toxin-encoding plasmids for the six MFDS strains. This pangenome is divided into the plasmids IncFI1, IncFIA, and phage. The graphical circular maps of plasmids are generated using GView.

**Figure 4 ijms-24-12729-f004:**
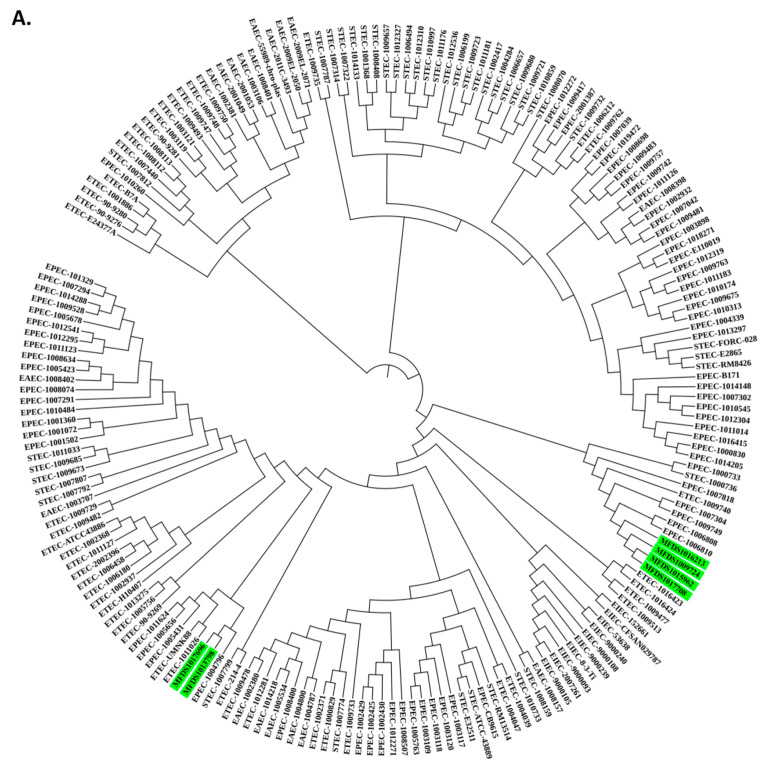
Phylogenetic and population structure analysis of hybrid EPEC/ETEC strains. (**A**) The green bars represent the six EPEC/ETEC hybrid strains that are closely related to specific EPEC strains; (**B**) Sequence clusters (1 to 5) are indicated in the outer colored dot, which are further divided into 27 lineages (inner ring). The green bar represents 6 EPEC/ETEC hybrid strains isolated identified in this study. Among the identified strains, six hybrid strains closely related to EPEC were divided into two groups (two level 1, two level 2).

**Table 1 ijms-24-12729-t001:** Summarized characteristics of hybrid EPEC/ETEC strains.

Strain Name	Collection Date	Geographic Location	Isolation Source	Coverage	Contigs	Size (bp)	GC (%)	CDSs	rRNA	tRNA	Accession No.
MFDS1009724	6 December 2017	Incheon	cow feces	251	3	5,132,531	50.7	5120	22	93	JARWBP000000000
MFDS1013798	12 November 2019	Incheon	cow feces	328	4	5,343,239	50.8	5465	22	98	JARWBW000000000
MFDS1015962	9 July 2020	Gyeongsangbuk-do	beef	643	4	5,183,510	50.7	5196	22	93	JARWBS000000000
MFDS1016213	26 November 2020	Incheon	cow feces	343	4	5,152,661	50.8	5201	22	95	JARWBT000000000
MFDS1017696	29 October 2020	Jeollabuk-do	beef	254	7	5,434,752	50.9	5756	22	99	JARWBV000000000
MFDS1017708	26 November 2020	Jeollabuk-do	beef	377	4	5,238,653	50.7	5300	22	93	JARWBW000000000

**Table 2 ijms-24-12729-t002:** Serotype and sequence type of hybrid EPEC/ETEC strains.

Strain Name	Collection Date	Serotype		Sequence Type
MFDS1009724	6 December 2017	O153	H19	ST546
MFDS1013798	12 November 2019	O49	H10	ST785
MFDS1015962	9 July 2020	O153	H19	ST546
MFDS1016213	26 November 2020	O71	H19	ST546
MFDS1017696	29 October 2020	O49	H10	ST785
MFDS1017708	26 November 2020	O153	H19	ST546

**Table 3 ijms-24-12729-t003:** Genome characterization of heat-stable toxins-encoding plasmids.

Strain Name	Plasmid Type	Virulence Genes and Transporters	Phage Sequence Regions		
			Most Common Phage Name	Completeness	Virulence Gene & Transporter
MFDS1009724	IncFI1	*estA*, *nleH2*, *hlyA*, *hlyB*, *hlyC*, *hlyD*, *espP*, Phage endopeptidase Rz, Hok/Gef-like protein	Stx2_c_1717	intact	Hok/Gef-like protein
			Entero_BP_4795	intact	*espP*, Phage endopeptidase Rz
MFDS1013798	IncFI1	*estA*, *nleH2*, *hlyA*, *hlyB*, *hlyC*, *hlyD*, *espP*, Phage endopeptidase Rz, Phage holin, Hok/Gef-like protein	Stx2_c_1717	intact	*espC*
			Salmon_SJ46	intact	*espP*
			Shigel_Sf6	intact	Phage endopeptidase Rz, Phage holin
MFDS1015962	IncFI1	*estA*, *nleH2*, *hlyA*, *hlyB*, *hlyC*, *hlyD*, *espP*, Phage endopeptidase Rz, Hok/Gef-like protein	Stx2_c_1717	intact	Hok/Gef-like protein
			Entero_BP_4795	intact	Phage endopeptidase Rz
MFDS1016213	IncFIA /IncFI1	*estA*, *nleH2*, *espC*, Phage endopeptidase Rz, Hok/Gef-like protein	Stx2_c_1717	intact	*espC*
			Entero_BP_4795	intact	-
			Escher_RCS47	intact	*nleH2*, Phage endopeptidase Rz
MFDS1017696	IncFI1	*estA*, *nleH2*, *hlyA*, *hlyB*, *hlyC*, *hlyD*, *espP*, Phage endopeptidase Rz, Hok/Gef-like protein	Stx2_c_1717	intact	*espP*
			Entero_BP_4795	intact	Phage endopeptidase Rz
MFDS1017708	IncFI1	*estA*, *nleH2*, *hlyA*, *hlyB*, *hlyC*, *hlyD*, *espP*, Phage endopeptidase Rz, Hok/Gef-like protein	Entero_BP_4795	intact	*espP*, Phage endopeptidase Rz

## Data Availability

Sequence data have been submitted to the publicly accessible NCBI archives (https://ncbi.nlm.nih.gov (accessed on 15 April 2023)), including GenBank, under the accession numbers listed in Table 1.

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
