# Peer review of "Molecular and Genomic Analysis of the Virulence Factors and Potential Transmission of Hybrid Enteropathogenic and Enterotoxigenic Escherichia coli (EPEC/ETEC) Strains Isolated in South Korea"

_ijms, 2023, doi:10.3390/ijms241612729_

Round 1

Reviewer 1 Report

an interesting study on molecular biology of EPEC/ETEC hybrid strains of DEC.

Introduction - nicely written paragraph - no remarks

Results - in-depth analysis of genome assembly, virulence factors genes, serotyping and sequence types of hybrid EC, heat-stable Toxin-encoding Plasmids and phylogenetic analysis

the results are presented nicely and properly visualized

Discussion and conclusion - authors elegantly incorporate their results in the literature on the subject and substantiate the conclusion that those hybrid strains can impose a significant danger for the public health and those types of analysis should be an integral part of epidemiological surveillance  

Author Response

an interesting study on molecular biology of EPEC/ETEC hybrid strains of DEC.

Introduction - nicely written paragraph - no remarks

Results - in-depth analysis of genome assembly, virulence factors genes, serotyping and sequence types of hybrid EC, heat-stable Toxin-encoding Plasmids and phylogenetic analysis

the results are presented nicely and properly visualized

Discussion and conclusion - authors elegantly incorporate their results in the literature on the subject and substantiate the conclusion that those hybrid strains can impose a significant danger for the public health and those types of analysis should be an integral part of epidemiological surveillance  

Response: Thank you for your comments.

Reviewer 2 Report

Dear Authors,

Congratulations on such massive work done!

The presented study is a very comprehensive research which first involved screening of huge number of strains (over 1,000), followed by a detailed analysis of six predetermined hybrid strains.

My only remarks involve redrafting the final section of Introduction. In details:

81-89: I assume that this section was to specify the aim of this study. However, in my opinion, it just describes what was done and does not provide a specific, scientific (or utilitarian) aim.

87-88: the sentence in these lines need correction. Obviously there are some crucial words missing.

The figures you presented provide abundance of data, therefore I understand their complexity. However, before submitting the final version of your manuscript, I suggest that you focus on providing them in high qualty large files that the readers would be able to open in another window.

Apart from that - I have no remarks.

The quality of English language is generally fine. Some parts of the manuscript read like there were some words missing (e.g. my comment on the final fragment of the Introduction: 87-88: the sentence in these lines need correction. Obviously there are some crucial words missing.)

However, having regard to the fact that all MDPI manuscripts undergo English editing before publication, I think that the mistakes that I noticed are just minor.

Author Response

Congratulations on such massive work done!

The presented study is a very comprehensive research which first involved screening of huge number of strains (over 1,000), followed by a detailed analysis of six predetermined hybrid strains.

My only remarks involve redrafting the final section of Introduction. In details:

81-89: I assume that this section was to specify the aim of this study. However, in my opinion, it just describes what was done and does not provide a specific, scientific (or utilitarian) aim.

Response: As you recommended, we revised the sentence in lines 82-94 as follows:

Lines 82-94: Few studies have reported genome-based characteristic analysis of EPEC/ETEC hybrid strains isolated in South Korea. In this study, we aimed to investigate the genomes of EPEC/ETEC hybrid strains to identify virulence factors from both EPEC and ETEC pathovars. We also figured out where the discovered strains fit in a large group of pathogenic E. coli strains that included all of the major pathotypes. Real-time polymerase chain reaction (PCR) and whole-genome sequencing (WGS) were used for investigation into the molecular features of these strains. To determine the evolutionary locations of these hybrids, a phylogenetic analysis was carried out. Genome-wide phylogenetic analysis revealed that these hybrids were closely related to certain EPEC strains. Through genome-based characterization, we confirmed that the EPEC/ETEC hybrid strains are likely EPEC strains that have acquired ETEC virulence genes via plasmids. On the basis of our research, we explored the possible effects that the hybrid E. coli strains detected may have on the health of the general public.

87-88: the sentence in these lines need correction. Obviously there are some crucial words missing.

Response: As you recommended, we revised the sentence in lines 89-92 as follows:

Lines 89-92: Genome-wide phylogenetic analysis revealed that these hybrids were closely related to certain EPEC strains. Through genome-based characterization, we confirmed that the EPEC/ETEC hybrid strains are likely EPEC strains that have acquired ETEC virulence genes via plasmids.

The figures you presented provide abundance of data, therefore I understand their complexity. However, before submitting the final version of your manuscript, I suggest that you focus on providing them in high qualty large files that the readers would be able to open in another window.

Response: As you recommended, we improved the resolution of figures and attached high quality large files